# LOQA: LEARNING WITH OPPONENT Q-LEARNING AWARENESS

**Milad Aghajohari**\*, **Juan Agustin Duque**\*, **Tim Cooijmans, Aaron Courville**
University of Montreal & Mila
`firstname.lastname`@umontreal.ca

## ABSTRACT

In various real-world scenarios, interactions among agents often resemble the dynamics of general-sum games, where each agent strives to optimize its own utility. Despite the ubiquitous relevance of such settings, decentralized machine learning algorithms have struggled to find equilibria that maximize individual utility while preserving social welfare. In this paper we introduce Learning with Opponent Q-Learning Awareness (LOQA), a novel, decentralized reinforcement learning algorithm tailored to optimizing an agent's individual utility while fostering cooperation among adversaries in partially competitive environments. LOQA assumes the opponent samples actions proportionally to their action-value function Q. Experimental results demonstrate the effectiveness of LOQA at achieving state-of-the-art performance in benchmark scenarios such as the Iterated Prisoner's Dilemma and the Coin Game. LOQA achieves these outcomes with a significantly reduced computational footprint, making it a promising approach for practical multi-agent applications.

## 1  INTRODUCTION

A major difficulty in reinforcement learning (RL) and multi-agent reinforcement learning (MARL) is the non-stationary nature of the environment, where the outcome of each agent is determined not only by their own actions but also those of other players von Neumann (1928). This difficulty often results in the failure of traditional algorithms converging to desirable solutions. In the context of general-sum games, independent RL agents often converge to sub-optimal solutions in the Pareto sense, when each of them seeks to optimize their own utility Foerster et al. (2018b). This situation draws parallels with many real-world scenarios, in which individuals pursuing their own selfish interests leads them to a worse outcome than cooperating with others. Thus one of the objectives of MARL research must be to develop decentralized agents that are able to cooperate while avoiding being exploited in partially competitive settings. We call this reciprocity-based cooperation.

Previous work has resulted in algorithms that train reciprocity-based cooperative agents by differentiating through the opponent's learning step (Foerster et al., 2018b; Letcher et al., 2021; Zhao et al., 2022; Willi et al., 2022) or by modeling opponent shaping as a meta-game in the space of agent policies (Al-Shedivat et al., 2018; Kim et al., 2021; Lu et al., 2022; Cooijmans et al., 2023). However, both of these approaches have important drawbacks with respect to computational efficiency. On one hand, differentiating through even just a few of the opponent's learning steps, can only be done sequentially and requires building large computation graphs. This is computationally costly when dealing with complex opponent policies. On the other hand, meta-learning defines the problem as a meta-state over the product space of policies of the agent and opponent, and learns a meta-policy that maps from the meta-state to the agent's updated policy. The complexity of the problem then scales with the policy parameterization which is usually a neural network with many parameters.

In this paper we introduce Learning with Opponent Q-Learning Awareness (LOQA), which stands because it avoids computing gradients w.r.t. optimization steps or learning the dynamics of a meta-game, resulting in significantly improved computational efficiency. LOQA performs opponent shaping by assuming that the opponent's behavior is guided by an internal action-value function $Q$. This assumption allows LOQA agents to build a model of the opponent policy that can be shaped by in-

fluencing its returns for different actions. Controlling the return by differentiating through stochastic objectives is a key idea in RL and can be done using the REINFORCE estimator Williams (1992).

## 2 BACKGROUND

We consider general-sum, $n$-player, Markov games, also referred as stochastic games Shapley (1953). Markov games are defined by a tuple $\mathcal{M} = (N, \mathcal{S}, \mathcal{A}, P, \mathcal{R}, \gamma)$ where $\mathcal{S}$ denotes the state space, $\mathcal{A} := \mathcal{A}^1 \times \ldots \times \mathcal{A}^n$, is the joint action space of all players, $P : \mathcal{S} \times \mathcal{A} \to \Delta(\mathcal{S})$, defines a mapping from every state and joint action to a probability distribution over states, $\mathcal{R} = \{r^1, \ldots, r^n\}$ is the set of reward functions where each $r^i : \mathcal{S} \times \mathcal{A} \to \mathbb{R}$ maps every state and joint action to a scalar return and $\gamma \in [0, 1]$ is the discount factor.

We use the notation and definitions for standard RL algorithms of Agarwal et al. (2021). Consider two agents, 1 (agent) and 2 (opponent) that interact in an environment with neural network policies $\pi^1 := \pi(\cdot|\cdot; \theta^1)$, $\pi^2 := \pi(\cdot|\cdot; \theta^2)$ parameterized by $\theta^1$ and $\theta^2$ respectively. We denote $\tau$ to be a trajectory with initial state distribution $\mu$ and probability measure $\Pr_\mu^{\pi^1, \pi^2}$ given by

$$\Pr_\mu^{\pi^1, \pi^2}(\tau) = \mu(s_0)\pi^1(a_0|s_0)\pi^2(b_0|s_0)P(s_1|s_0, a_0, b_0)\ldots$$

here $b \in \mathcal{A}^2$ denotes the action of the opponent. In multi-agent reinforcement learning, each agent seeks to optimize their expected discounted return $R$, for the agent this is given by:

$$V^1(\mu) := \mathbb{E}_{\tau \sim \Pr_\mu^{\pi^1, \pi^2}}\left[R^1(\tau)\right] = \mathbb{E}_{\tau \sim \Pr_\mu^{\pi^1, \pi^2}}\left[\sum_{t=0}^{\infty} \gamma^t r^1(s_t, a_t, b_t)\right]$$

The key observation is that under the definitions above, $V^1$ is dependent on the policy of the opponent through the reward function $r^1(s_t, a_t, b_t)$. $V^1$ is thus differentiable with respect to the parameters of the opponent via the REINFORCE estimator (Williams, 1992)

$$
\begin{aligned}
\nabla_{\theta^2} V^1(\mu) &= \mathbb{E}_{\tau \sim \Pr_\mu^{\pi^1, \pi^2}}\left[R^1(\tau) \sum_{t=0}^{\infty} \nabla_{\theta^2}\log \pi^2(b_t|s_t)\right] \\
&= \mathbb{E}_{\tau \sim \Pr_\mu^{\pi^1, \pi^2}}\left[\sum_{t=0}^{\infty} \gamma^t r^1(s_t, a_t, b_t) \sum_{k<t} \nabla_{\theta^2}\log \pi^2(b_k|s_k)\right]
\end{aligned}
\tag{1}
$$

we will rely on this observation to influence the opponent's $Q$ value and incentivize reciprocity-based cooperation in partially competitive environments.

## 3 RELATED WORK

### 3.1 OPPONENT SHAPING

Learning with opponent learning awareness (LOLA), (Foerster et al., 2018b) introduces the concept of opponent shaping, i.e. the idea of steering the other agent throughout its learning process. LOLA assumes that the opponent is a naive learner and attempts to shape it by considering one step in its optimization process. Rather than optimizing the value under the current policies at the current iteration $i$, $V^1(\theta_i^1, \theta_i^2)$, LOLA optimizes $V^1(\theta_i^1, \theta_i^2 + \Delta\theta_i^2)$ where $\Delta\theta_i^2$ is a learning step of the opponent. $\Delta\theta_i^2$ is as a function that depends on the agent's parameters and that is thus differentiable with respect to $\theta^1$. Since the derivative of function $V^1(\theta_i^1, \theta_i^2 + \Delta\theta_i^2)$ is difficult to compute, the authors consider the surrogate value given by its first order Taylor expansion. POLA, (Zhao et al., 2022) builds an idealized version LOLA that, unlike its predecessor, is invariant to the parameterization of the policy. In a similar fashion to proximal policy optimization (PPO) (Schulman et al., 2017), each agent increases the probability of actions that increase their expected return, while trying to minimize the $l_2$ distance between the updated policy and the old policy. This combined objective of maximizing the return and minimizing the $l_2$ distance in policy space is equivalent to the Proximal Point method, hence the name Proximal-LOLA or POLA. Other modifications to the original LOLA algorithm attempt to resolve its shortcomings. Consistent learning with opponent learning awareness (COLA), (Willi et al., 2022) attempts to resolve the inherent inconsistency of LOLA assuming

| Agent \ Opponent | Cooperate | Defect |
|---|---|---|
| Cooperate | $(-1, -1)$ | $(-3, \ 0)$ |
| Defect | $( \ 0, -3)$ | $(-2, -2)$ |

Table 1: Payoff (or reward) matrix for the Prisoner's Dilemma game, where the numbered pairs corresponds to the payoffs of the Agent and the Opponent respectively.

that the other agent is a naive learner instead of another LOLA agent. Stable opponent shaping (SOS), (Letcher et al., 2021) introduces an interpolation between LOLA and a more stable variant called *look ahead*, which has strong theoretical convergence guarantees.

## 3.2 META LEARNING

Other methods have been used to generate agents that have near optimal behavior in social dilemmas. First used by Al-Shedivat et al. (2018) for this setting, meta learning redefines the MARL problem as a meta-game in the space of policy parameters in an attempt to deal with the non-stationary nature of the environment. In this meta-game, the meta-state is the joint policy, the meta-reward is the return on the underlying game, and a meta-action is a change to the inner policy (i.e. the policy in the original game). Model free opponent shaping (M-FOS) (Lu et al., 2022) applies policy gradient methods to this meta-game to find a strong meta-policy. Meta-Value Learning (Cooijmans et al., 2023) applies value learning to model the long-term effects of policy changes, and uses the gradient of the value as an improvement direction.

## 4 SOCIAL DILEMMAS

Social dilemmas are a type of decision problem where each party's miopic efforts to maximize their own benefit lead to a less favorable outcome compared to when all parties cooperate. Designed primarily as thought experiments, these dilemmas demonstrate the trade-offs that are often inherent in multi-agent decision-making scenarios. As such, they have been used to model real-life situations in diverse fields such as economics, ecology and policy making. One example of such a decision problem is the famous Iterated Prisoner's Dilemma (IPD).

**The Prisoner's Dilemma (PD)** PD is a game in which each of the two agents or prisoners must decide to either cooperate with one another or defect. The dilemma the prisoners face originates from the reward structure, given in Table 1. With with reward structure, a rational agent will choose to defect no matter what the other agent chooses. As a result, both agents become locked in the defect-defect Nash equilibrium, even though they would achieve greater utility by both choosing to cooperate.

**Iterated Prisoner's Dilemma (IPD)** As the name implies IPD is simply an (infinitely) repeated version of the Prisoner's Dilemma. Unlike standard PD, IPD offers some hope for rational cooperative behaviour. Originally popularized by Axelrod (1980), the IPD has been used to model many hypothetical and real-world scenarios. It has also become a popular test-bed for MARL algorithms attempting to achieve reciprocity based cooperation policies. A simple but effective strategy in the IPD is *tit-for-tat* (TFT), which consists in cooperating at the first turn and copying the opponent's action thereafter.

**The Coin Game** Initially described in (Lerer & Peysakhovich, 2018), the Coin Game is a grid world environment in which two agents take turns taking coins. At the beginning of each episode a coin of a particular color (red or blue), corresponding to that of one of the two agents, spawns at a random location in the grid. Agents are rewarded for any coin taken, but are punished if the other agent takes the coin corresponding to their color. The reward structure of the Coin Game is designed to incentivize cooperation between agents, as each one would be better off if both take only the coin corresponding to their color. In such way the Coin Game mirrors the IPD, therefore policies that *cooperate reciprocally* are highly desirable as they achieve better individual and social outcomes.

However unlike IPD, the Coin Game is embedded in a non-trivial environment and requires non-trivial policy models. It can be seen as an extension of IPD towards more complex and realistic scenarios.

## 5 METHOD DESCRIPTION

Intuitively, a change in either the agent's or the opponent's policy results in a change in the probability measure over the trajectories that are observed when both of them interact in an environment. Since the value function of the *opponent* is an expectation over said probability measure, it is controllable by the *agent's* policy (and vice versa). LOQA leverages this observation to exert influence over the policy that the opponent will learn.

As an illustration, consider an instance of the IPD game where a LOQA agent and the opponent are initialized to be random agents, i.e. they samples actions from a uniform distribution. If the LOQA agent increases its probability of defection after the opponent defects, it implicitly decreases the action-value function of the opponent for defecting. The opponent will then learn this and reduce its probability of defecting. Similarly, if the LOQA agent cooperates after the opponent cooperates, it increases the action-value of cooperation for the opponent. In response, the opponent will learn to cooperate more. This reciprocity-based cooperative behavior is the structure behind tit-for-tat.

### 5.1 MODELING THE OPPONENT'S POLICY

Let $\pi^1(b|s) := \pi(b|s; \theta^1)$ refer to the policy of the agent and $\pi^2(b|s) := \pi(b|s; \theta^2)$ refer to the policy of the opponent, which are neural networks with parameters $\theta^1$ and $\theta^2$. Similarly, $Q^2(s, b) := Q(s, b; \phi^2)$ denotes the action-value function of the opponent, which is a neural network with parameters $\phi^2$.

LOQA relies on a key assumption about the opponent's policy. Similar to Soft Actor Critic (SAC) (Haarnoja et al., 2018), the assumption is that the opponents' actions are sampled from a distribution that is proportional to its action-value function $Q^2(\cdot)$. More formally, at time $t$, we can write this assumption as

$$\pi^2(b_t|s_t) \approx \frac{\exp(Q^2(s_t, b_t))}{\sum_{b'} \exp(Q^2(s_t, b'))}$$

More specifically we approximate $Q^2$ with Monte Carlo rollouts $\mathcal{T}$ of length $T$, where every trajectory $\tau \in \mathcal{T}$, $\tau \sim \Pr_\mu^{\pi^1, \pi^2}$, starts at state $s_t$ with the opponent taking action $b_t$

$$\hat{Q}^2(s_t, b_t) = \mathbb{E}_{\tau \sim \mathcal{T}} \left[ R^2(\tau) | s = s_t, b = b_t \right]$$

$$= \frac{1}{|\mathcal{T}|} \sum_{\tau \in \mathcal{T}} \sum_{k=t}^{T} \gamma^{k-t} r^2(s_k, a_k, b_k)$$

where $r^2(s, a, b)$ denotes the opponent's reward at state $s$ after taking action $b$ and the opponent taking action $a$. This empirical expectation of the $Q$ function is controllable by the agent using the reinforce estimator

$$\nabla_{\theta^1} \hat{Q}^2(s_t, b_t) = \mathbb{E}_{\tau \sim \mathcal{T}} \left[ \sum_{k=t+1}^{T} \gamma^{k-t} r^2(s_k, a_k, b_k) \sum_{t < j < k} \nabla_{\theta^1} \log \pi^1(a_j|s_j) \right]$$

The opponent's policy evaluated at state $s_t$ can now be approximated using the Monte Carlo rollout estimate $\hat{Q}^2$ and the action-value function $Q^2$ as follows

$$\hat{\pi}^2(b_t|s_t) := \frac{\exp(\hat{Q}^2(s_t, b_t))}{\exp(\hat{Q}^2(s_t, b_t)) + \sum_{b' \neq b_t} \exp(Q^2(s_t, b'))} \tag{2}$$

Notice that we assume access to the opponent's real action-value function $Q^2$. To have a fully decentralized algorithm we can simply replace $Q^2$ with the agent's own estimate of the opponent's action-value function. We now integrate these ideas into the Actor-Critic formulation.

---

**Algorithm 1** LOQA

---

**Initialize:** Discount factor $\gamma$, agent action-value parameters $\phi^1$, target action-value parameters $\phi^1_{\text{target}}$, actor parameters $\theta^1$, opponent action-value parameters $\phi^2$, target action-value parameters $\phi^2_{\text{target}}$, actor parameters $\theta^2$
**for** iteration$= 1, 2, \ldots$ **do**
    Run policies $\pi^1$ and $\pi^2$ for $T$ timesteps in environment and collect trajectory $\tau$
    $L^1_Q \leftarrow 0, L^2_Q \leftarrow 0$
    **for** $t = 1, 2, \ldots, T-1$ **do**
        $L^1_Q \leftarrow L^1_Q + \text{HUBER\_LOSS}(r_t + \gamma Q^1_{\text{target}}(s_{t+1}, a_{t+1}) - Q^1(s_t, a_t))$
        $L^2_Q \leftarrow L^2_Q + \text{HUBER\_LOSS}(r_t + \gamma Q^2_{\text{target}}(s_{t+1}, b_{t+1}) - Q^2(s_t, a_t))$
    **end for**
    Optimize $L^1_Q$ w.r.t. $\phi^1$ and $L^2_Q$ w.r.t. $\phi^2$ with optimizer of choice
    Compute advantage estimates $\{A^1_1, \ldots, A^1_T\}, \{A^2_1, \ldots, A^2_T\}$
    $L^1_a \leftarrow \text{LOQA\_ACTOR\_LOSS}(\tau, \pi^1, \gamma, \{A^1_1, \ldots, A^1_T\})$
    $L^2_a \leftarrow \text{LOQA\_ACTOR\_LOSS}(\tau, \pi^2, \gamma, \{A^2_1, \ldots, A^2_T\})$
    Optimize $L^1_a$ w.r.t. $\theta^1$ and $L^2_a$ w.r.t. $\theta^2$ with optimizer of choice
**end for**

---

**Algorithm 2** LOQA\_ACTOR\_LOSS

---

**Input:** Trajectory $\tau$ of length $T$, actor policy $\pi^i$, opponent action-value function $Q^{-i}$, discount factor $\gamma$, advantages $\{A^i_1, \ldots, A^i_T\}$
$L_a \leftarrow 0$
**for** $t = 1, 2, \ldots, T-1$ **do**
    $\hat{Q}^{-i}(s_t, b_t) \leftarrow \sum_{k=t}^{T} \gamma^{k-t} r^{-i}(s_k, a_k, b_k)$         $\triangleright$ $r^{-i}$ made differentiable using DiCE
    Compute $\hat{\pi}^{-i}$ using $\hat{Q}^{-i}(s_t, b_t)$ and $Q^{-i}(s_t, b_t)$ according to equation (2)
    $L_a \leftarrow L_a + A^i_t \left[ \log \pi^i(a_t|s_t) + \log \hat{\pi}^{-i}(b_t|s_t) \right]$
**end for**
**return:** $L_a$

---

## 5.2 OPPONENT SHAPING

In order to shape the opponent behavior, we factor in the opponent's policy approximation $\hat{\pi}^2$ as well as the agent's policy $\pi^1$ in the probability measure over trajectories. Adapting the original Actor-Critic formulation (Konda & Tsitsiklis, 2000) to the joint agent-opponent policy space we have:

$$\nabla_{\theta^1} V^1(\mu) = \mathbb{E}_{\tau \sim \text{Pr}^{\pi^1, \pi^2}_\mu} \left[ \sum_{t=0}^{T} A^1(s_t, a_t, b_t) \nabla_{\theta^1} \left( \log \pi^1(a_t|s_t) + \underbrace{\log \pi^2(b_t|s_t)}_{=0} \right) \right]$$

where $A^1(s_t, a_t, b_t)$ is the advantage of the first agent, and $\pi^2$ is constant w.r.t. $\theta_1$. LOQA approximates the opponent's policy using Equation 2. This approximated policy is differentiable with respect to agent parameters since it is computed based on the opponent's action-value, which is also differentiable (see Equation 1). Consequently, a second term emerges in LOQA's update

$$\nabla_{\theta^1} V^1(\mu) = \mathbb{E}_{\tau \sim \text{Pr}^{\pi^1, \pi^2}_\mu} \left[ \sum_{t=0}^{T} A^1(s_t, a_t, b_t) \nabla_{\theta^1} \left( \log \pi^1(a_t|s_t) + \log \hat{\pi}^2(b_t|s_t) \right) \right] \quad (3)$$

The first log term comes from the Actor-Critic update and the second log term is a shaping component that pushes the opponent's return in a beneficial direction for the agent (in policy space). This second term comes from the assumption that the opponent's policy can be influenced by the agent's parameters. For a derivation refer to section F in the appendix.

In practice we use DiCE (Foerster et al., 2018a) and loaded-DiCE (Farquhar et al., 2019) on the action-value estimate $\hat{Q}^2$ to compute the gradient $\nabla_{\theta^1} \log \hat{\pi}^2$ and reduce its variance. Also, the current trajectory $\tau$ itself is used for $\hat{Q}^2$ estimation. (See appendix B)

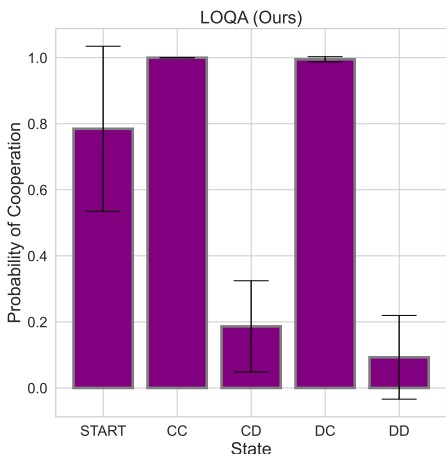

Figure 1: Probability of cooperation of a sigmoid LOQA agent at each possible state in the one-step history IPD after 7000 training iterations. LOQA agents' resulting policy is similar to tit-for-tat, a policy that cooperates at the first step and copies the previous action of the opponent at subsequent time-steps.

### 5.3 SELF PLAY AND REPLAY BUFFER OF PREVIOUS AGENTS

Because the environments are symmetric we can use *self-play* to train a single LOQA agent against itself. We also maintain a replay buffer and, for each optimization step (which requires generating environment rollouts), we sample uniformly from previously encountered agents. This increases the diversity of the opponents the agent faces in the training. The replay buffer has a certain capacity and receives a new agent every $n$th iteration where $n$ is a hyperparameter.

## 6 EXPERIMENTS

We consider two general-sum environments to evaluate LOQA against the current state-of-the-art, namely, the Iterated Prisoner's Dilemma (IPD) and the Coin Game. We compare with POLA and M-FOS, the only methods to the best of our knowledge that generate reciprocity-based cooperative policies in the Coin Game.

### 6.1 ITERATED PRISONER'S DILEMMA

We train an agent consisting of a sigmoid over logits for each possible state in the one-step history IPD. There are 5 possible states in this configuration, namely START (the starting state), CC, CD, DC and DD where C stands for cooperation and D stands for defection. The training is done for 4500 iterations (approximately 15 minutes on an Nvidia A100 gpu) using a batch size of 2048. We empirically observe that a LOQA agent is able to reach a tit-for-tat like policy as shown by looking at the probability of cooperation at each state. We believe that the probabilities are not fully saturated for two reasons. First, the normalization over the action-values in the opponent's policy approximation makes it numerically impossible to reach a probability of one for either action. Second, we observed that after some time the trajectories become homogeneous "always cooperate" trajectories that ultimately degenerate the quality of the resulting policy by making it less likely to retaliate after a defection by the opponent.

### 6.2 COIN GAME

Like (Zhao et al., 2022), we use a GRU policy that has access to the current observation of the game plus both agents' actions in the previous turn to train a LOQA agent in the Coin Game. We run trajectories of length 50 with a discount factor $\gamma = 0.7$ in parallel with a batch size of 8192. For evaluation we run 10 seeds of fully trained agents for 50 episodes in a league that involves other

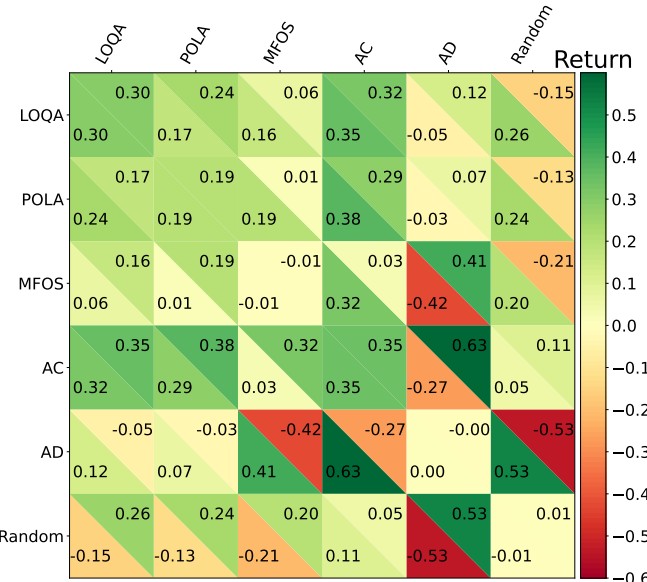

Figure 2: Average rewards after evaluating 10 fully trained LOQA, POLA, and M-FOS seeds against different agents in a 3x3 sized Coin Game lasting 50 episodes. AC for always Cooperate, AD for always defect. Notice that a fully cooperative agent achieves an average reward of 0.35 against itself. LOQA is able to generate a policy that demonstrates reciprocity-based cooperation.

agents and plot the time-averaged reward. The results are shown in Figure 2. We also experimented with ablations of LOQA that either removed self-play during training or removed the replay buffer of past policy weights to train against. These two ablations showed that these two elements, although not essential, improve the performance of LOQA agents in the Coin Game by making them more cooperative with themselves and less exploitable by always-defect agents. For details refer to the Appendix section C.

In Figure 2, we observe that LOQA agents are able to cooperate with themselves as indicated by the high average reward of 0.3 which is close to the 0.35 of an always cooperate agent against itself. LOQA agents are able to achieve high social welfare without being exploited by an always defect agent as they achieve an average reward of -0.05, which is comparable to POLA's own -0.03. More importantly our agents are fully trained after only 2 hours of compute time in an Nvidia A100 gpu, compared to the 8 hours of training it takes POLA to achieve the results shown in Figure 2. It should be noted that as compared to IPD, in the Coin Game, cooperation and defection consist of sequences of actions, therefore an agent must learn to take coins before learning whether they should cooperate with their opponent or not. We also consider full histories as opposed to one-step, making the state space significantly larger.

## 6.3 SCALABILITY EXPERIMENTS

In this section, we test LOQA, POLA, and M-FOS on larger grid sizes in the Coin Game to evaluate their scalability. As grid size increases, the average distance between the agents and the coin also grows This added complexity challenges the learning of reciprocal cooperative behavior. For example, when the opponent takes the agent's coin, the agent must learn to take multiple steps to retaliate. This behavior is less likely to be discovered by random actions on larger grids. Our experiments with different grid sizes illustrate LOQA's scalability properties compared to POLA and M-FOS.

In assessing the performance of the agents, we consider two metrics: the achievement of a predetermined performance threshold and the time taken to reach this threshold. For the latter, it is critical to acknowledge that the conceptualization of a 'step' differs between POLA and LOQA. The steps in POLA encompass numerous inner optimization steps, rendering a direct comparison of performance

| Threshold | Normalized Return against Each Other | Normalized Return against Always Defect |
|---|---|---|
| Weak | $\geq 0.05$ | $\geq -1.2$ |
| Medium | $\geq 0.1$ | $\geq -0.5$ |
| Strong | $\geq 0.2$ | $\geq -0.2$ |

Table 2: Thresholds based on two main criteria of a reciprocity-based cooperative policy. The weak and medium thresholds are designed such that all agents pass them, while the strong threshold represents good performance.

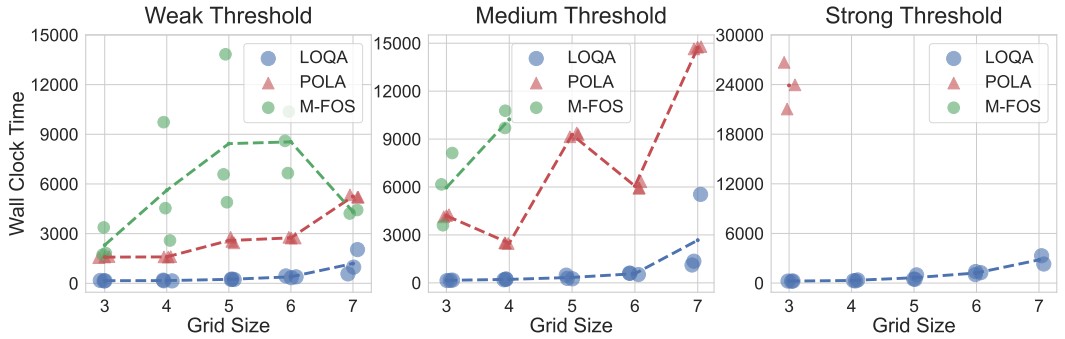

Figure 3: Wall clock time vs grid size for three seeds of LOQA, POLA, and M-FOS on reaching different thresholds. Each data point indicates the first time its corresponding seed passed a certain threshold. The wall clock time is measured in seconds. Red triangles indicate LOQA's performance while blue circles visualize LOQA's performance. Dashed lines pass through the average time for runs that passed for their respective algorithm.

per step inconclusive. To facilitate an equitable comparison, we employ the wall clock time of both algorithms, executed under identical computational configurations (GPU, memory, etc.).

The thresholds are defined based on two principal criteria pertaining to a reciprocity-based cooperative policy. Firstly, we evaluate the agent's return against an "Always Defect" opponent; this serves to test the cessation of cooperation following a lack of reciprocation from the opponent. Secondly, we consider the agents' return against each other, which serves as a measure of cooperative behavior. This dual-threshold approach is pragmatic as it discerns the distinctive behaviors; an 'Always Defect' agent meets the first threshold but fails the second, whereas an 'Always Cooperate' agent satisfies the second but not the first. However, a policy resembling 'Tit-for-Tat' satisfies both thresholds. Furthermore, as the grid size grows the average returns of agents per step decrease since it takes longer to reach the coin. Therefore, it is crucial to ensure our thresholds are consistent when evaluating for different grid sizes.

We normalize the returns to make our thresholds standard over all grid sizes. Specifically, we multiply the return by the maximum distance possible between the agent and the coin for a given grid size. Since the grid is wrapped, the normalization value $N$ is given by the Manhattan distance between the center of the grid and one of the corners. We call this the normalized return and we calculate our thresholds based on it. We have three thresholds for our evaluation. The weak and medium thresholds are designed so that all the algorithms are able to reach them. However, as LOQA reaches much higher return on large grid sizes as compared to POLA and M-FOS, we set the strong threshold to a high performance. The specification of the thresholds values is shown in Table 2.

The results of our experiments on larger grid sizes are illustrated in Figure 3. All LOQA runs meet the strong threshold for grid sizes up to 6, but at a grid size of 7, one run falls short of the strong threshold. In contrast, every POLA and M-FOS run fail to reach the strong threshold for grid sizes above 3. For further details on the training curves from these experiments, please see the appendix section D. Figure 6 also provides the evaluation metrics for each algorithm, detailing their behaviors. Additionally, LOQA consistently achieves each threshold substantially faster, by at least one order of magnitude because of its lower time and memory complexity.

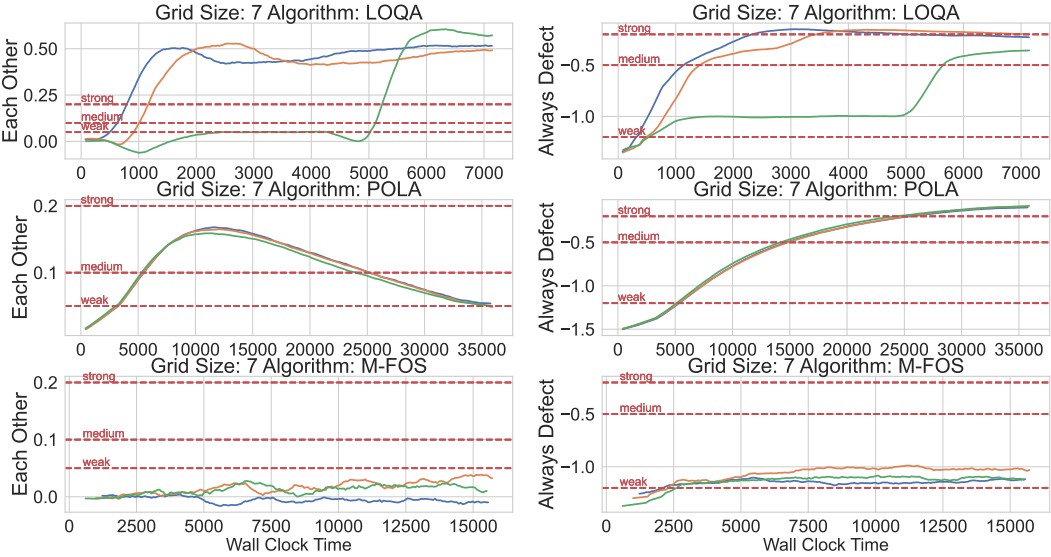

Figure 4: Training curves for 3 seeds of POLA, LOQA, and M-FOS on the two evaluation metrics for a 7x7 grid size: Normalized return vs. themselves (Self) and vs. always defect (AD). The wall clock time is measured in seconds. Note that the range of the x-axis is different for each algorithm as POLA and M-FOS need more time.

The complexity of LOQA is equivalent to the calculation of a REINFORCE estimator, as is standard in RL. Unlike POLA, LOQA does not involve calculations for opponent optimization steps nor does it differentiate through a computational graph of said optimizations during each training step, giving LOQA greater computationoal efficiency compared to POLA. Additionally, the absence of a second-order gradient in LOQA reduces the variance of its gradient estimators.

The model of opponent learning in POLA is restricted to a limited number of optimization steps. In scenarios with increased complexity and varying opponent policies, additional steps may be necessary to accurately represent the opponent's learning. This increase necessitates extended runtimes and increased memory allocation for storing the computational graphs required for differentiation, positioning LOQA as more efficient and economical in memory usage.

## 7 LIMITATIONS

LOQA is primarily limited by the assumption that the other player acts accordingly to an inner action-value function. As such, it is unable to shape other opponents that do not necessarily follow this assumption. In a similar way, LOQA agents are designed for environments with discrete action spaces. Future work could explore relaxations that allow LOQA agents to shape other types of agents and learn in continuous action spaces.

## 8 CONCLUSION

In this paper we have introduced LOQA, a decentralized reinforcement learning algorithm that is able to learn reciprocity-based cooperation in general sum environments at a lower computational cost than its predecessors. To do so, LOQA agents rely on the observation that their opponent's action-value function is controllable, and the assumption that their opponent's policy closely mirrors it. As a result, LOQA agents are able to shape other LOQA agents by performing REINFORCE updates that can be efficiently computed in hindsight after collecting environment trajectories. This is especially advantageous as demonstrated in the experimental setup, where LOQA agents confidently outperform POLA and M-FOS agents in terms of optimality and efficiency in the Coin Game. Therefore, LOQA stands out as a promising algorithm for tackling more complex and empirically-grounded social dilemmas.

## 9 ACKNOWLEDGMENTS

The authors would like to thank Mila and Compute Canada for providing the computational resources used for this paper. We would like to thank Olexa Bilaniuk for his invaluable technical support throughout the project. We acknowledge the financial support of Hitachi Ltd, Aaron's CIFAR Canadian AI chair and Canada Research Chair in Learning Representations that Generalize Systematically. Special thanks to Shunichi Akatsuka for his insightful discussions. We would like to thank the JAX ecosystem Bradbury et al. (2018).

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

APPENDIX

## A  EXPERIMENTAL DETAILS

### A.1  IPD

**Game specification** We use a finite prisoner dilemma of game length 50. The hyperparameters of our training are indicated in the Table 3

| Hyperparameter | Value |
|---|---|
| Actor Model | logits |
| Advantage Estimation Method | TD0 |
| Gradient Clipping Mode | disabled |
| Entropy Regularization Beta | 0 |
| Actor Learning Rate | 0.001 |
| Q-value Learning Rate | 0.01 |
| Target EMA Gamma | 0.99 |
| Optimizer | adam |
| Batch Size | 2048 |
| Game Length | 50 |
| Epsilon Greedy | 0.2 |
| Reward Discount | 0.96 |
| Agent Replay Buffer Mode | disabled |
| Differentiable Opponent Method | n step |
| Differentiable Opponent N Step | 2 |

Table 3: List of hyperparameters used in the training run.

**Agent's architecture** The agent's policy is modeled via 5 logits. We take a sigmoid over these logits to indicate the probability of cooperation on each of Start, CC, CD, DC, DD states. As the critic, we use a GRU unit described in A.2.

**Advantage Estimation** In our experiments we used TD-0 for advantage estimation. We estimate the value for a state as $V(s_t) = \sum_{a_t} Q(s_t, a_t)\pi(a_t|s_t)$. Consequently, we use $r_t + \gamma V(s_{t+1}) - V(s_t)$ as the advantage.

### A.2  COIN GAME

**GPU and Running Time:** We run our Coin Game experiments for 15 minutes on a single A100 GPU with 80 Gigabytes of GPU memory and 20 Gigabytes of CPU memory. We choose the last state of the training as our checkpoint which is the state at the 6000 iteration.

**Agent's architecture** We use the same GRU module taken from POLA's codebase. It is a GRU with two dense layers before the GRU unit and a single linear layer after GRU's output. We use one for actor and one for our critic.

**Replay Buffer** We maintain a replay buffer of agents visited during the training. It has a capacity of storing 10000 agents. We also push a new agent every 10 step. During trainig we sample an agent from the replay buffer uniformly.

**Hyperparameters** In table 4 we show the hyperparameters we used for training the agent.

### A.3  SCALABILITY EXPERIMENTS

**GPU and Running Time:** In our scalability experiments we train the LOQA agents for a maximum of two hours on a single node that has one A100 GPU with 80 Gigabytes of GPU memory on a system with 20 Gigabytes of CPU memory. For POLA we use a longer runtime of 10 hours on the same node configuration as it takes longer to achieve reasonable performance.

**Hyperparameters:** For POLA and M-FOS we take the hyperparameters as suggested on their respetive github repository. We use those same hyperparameters for all the grid sizes. For LOQA,

| Hyperparameter | Value |
|---|---|
| Grid Size | 3 |
| Game Length | 50 |
| Advantage Estimation Method | TD0 |
| Gradient Clipping Method | Norm |
| Gradient Clipping Max Norm | 1 |
| Actor Optimizer | Adam |
| Actor Loss Entropy Beta | 0.1 |
| Actor Learning Rate | 0.001 |
| Actor Hidden Size | 128 |
| Actor Dense Layers Before GRU | 2 |
| Critic Estimation | Mean |
| Critic Optimizer | Adam |
| Critic Learning Rate | 0.01 |
| Critic Target Exponential Moving Average's Gamma | 0.99 |
| Critic Hidden Size | 64 |
| Critic Dense Layers Before GRU | 2 |
| Batch Size | 512 |
| Reward Discount | 0.96 |
| Agent Replay Buffer Mode | Enabled |
| Agent Replay Buffer Capacity | 10000 |
| Agent Replay Buffer Update Freq | 10 |
| Differentiable Opponent Method | Loaded-Dice |
| Differentiable Opponent Discount | 0.9 |

Table 4: List of hyperparameters used in the training run.

we use the same hyperparameters as we found to work suitably on the 3x3 board. Although, we increased the batch size of LOQA to be 2048 to ensure reduced variance. In our 3x3 experiments for running the league we reduced the batch size to 512 so we can run ten seeds faster.

# B  IMPLEMENTATION DETAILS

Using naive REINFORCE estimators to compute the gradient with respect to the differentiable action-value estimate of the opponent leads to a suboptimal performance of LOQA due to high variance. We instead use a loaded-DiCE (Farquhar et al., 2019) implemenation of the actor loss. The implementation algorithm is provided in 3.

# C  ABLATIONS

## C.1  SELF-PLAY ABLATIONS

We achieve the results presented in the main paper by training one agent against itself [1]. Figure 5 shows the performance of L-S (LOQA without Self Play) agents. The performance of L-S is close to the L (LOQA with Self Play) although, L is stronger in cooperating with itself.

## C.2  AGENT REPLAY BUFFER ABLATIONS

Inspired by (Mnih et al., 2013), we use a replay buffer with capacity to store 10000 agents. Every 10 training steps, we push a copy of the agent's parameters to the replay buffer. Figure 5 shows the performance of L-R (LOQA without Replay Buffer). The replay buffer gets updated slowly with new agents, thus, it decreases the variance of the training process.

---

[1]Note when we use the replay buffer of agents, we train the agent against a sampled agent from the replay buffer. The replay buffer is filled with the past versions of the same agent encountered in training.

---

**Algorithm 3** LOQA_ACTOR_LOSS with loaded-DiCE

---

**Input:** Trajectory $\tau$ of length $T$, actor policy $\pi^i$, opponent action-value function $Q^{-i}$, discount factor $\gamma$, action discount factor $\lambda$, advantages $\{A_1^i, \ldots, A_T^i\}$
$L_a \leftarrow 0$
**for** $t = 1, 2, \ldots, T - 1$ **do**
   $w \leftarrow 0$
   $\hat{Q}^{-i}(s_t, b_t) \leftarrow \sum_{k=t}^{T} \gamma^{k-t} r^{-i}(s_k, a_k, b_k)$
   **for** $k = 2, \ldots, T$ **do**
      $w \leftarrow \lambda \cdot w + \log\left(\pi^i(a_t|s_t)\right)$
      $v \leftarrow \lambda \cdot w - \log\left(\pi^i(a_t|s_t)\right)$
      $\hat{Q}^{-i}(s_t, b_t) \leftarrow \hat{Q}^{-i}(s_t, b_t) + A_k^i(f(w) - f(v))$
   **end for**
   Compute $\hat{\pi}^{-i}$ using $\hat{Q}^{-i}(s_t, b_t)$ and $Q^{-i}(s_t, b_t)$ according to equation (2)
   $L_a \leftarrow L_a + A_t^i \left[\log \pi^i(a_t|s_t) + \log \hat{\pi}^{-i}(b_t|s_t)\right]$
**end for**
**return:** $L_Q$

**function** $f(x)$
   **return:** $\exp(x - \text{STOP\_GRADIENT}(x))$
**end function**

---

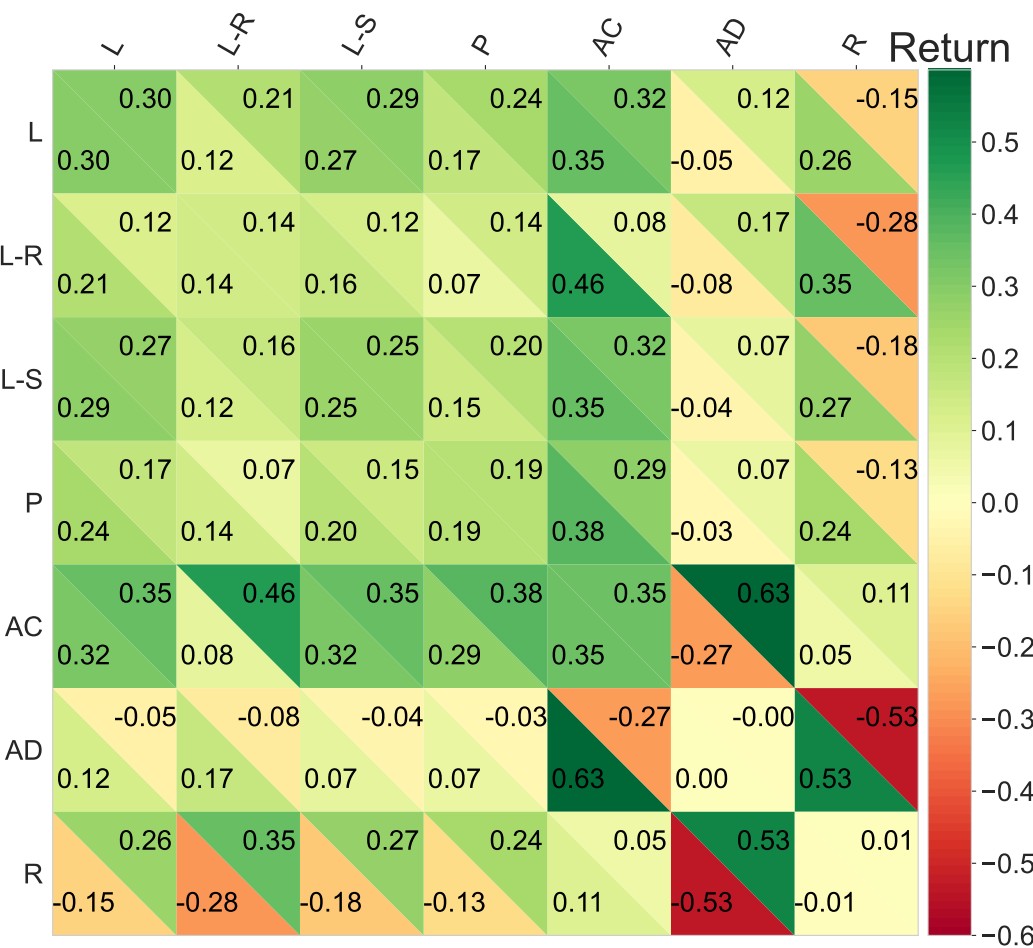

Figure 5: Average rewards after evaluating 10 fully trained LOQA and POLA seeds against different agents in a 3x3 sized Coin Game. We abbreviate L:LOQA, L-R:LOQA without replay buffer, L-S:for LOQA without self-play, P:POLA, AC:Always Cooperate, AD:Always defect and R:Random.

## D  SCALABILITY EXPERIMENTS

## E  LOQA'S ACTOR CRITIC UPDATE DERIVATION

In LOQA we assume that the trajectories are generated by policies $\pi^1$ and $\hat{\pi}^2$, which induces a different probability measure over trajectories

$$\mathrm{Pr}_{\mu}^{\pi^1,\hat{\pi}^2}(\tau) = \mu(s_0)\pi^1(a_0|s_0)\hat{\pi}^2(b_0|s_0)P(s_1|s_0,a_0,b_0)\dots$$

Our expected return must now be computed using this probability measure

$$\hat{V}^1(\mu) := \mathbb{E}_{\tau \sim \mathrm{Pr}_{\mu}^{\pi^1,\hat{\pi}^2}}\left[R^1(\tau)\right] = \mathbb{E}_{\tau \sim \mathrm{Pr}_{\mu}^{\pi^1,\hat{\pi}^2}}\left[\sum_{t=0}^{\infty}\gamma^t r^1(s_t,a_t,b_t)\right]$$

Therefore we can write

$$\nabla_{\theta^1}\hat{V}^1(\mu) = \nabla_{\theta^1}\sum_{\tau}R^1(\tau)\mathrm{Pr}_{\mu}^{\pi^1,\hat{\pi}^2}(\tau)$$

$$= \sum_{\tau}R^1(\tau)\mathrm{Pr}_{\mu}^{\pi^1,\hat{\pi}^2}(\tau)\nabla_{\theta^1}\log\left(\mathrm{Pr}_{\mu}^{\pi^1,\hat{\pi}^2}(\tau)\right)$$

$$= \sum_{\tau}R^1(\tau)\mathrm{Pr}_{\mu}^{\pi^1,\hat{\pi}^2}(\tau)\nabla_{\theta^1}\log\left(\prod_{t=0}^{\infty}\pi^1(a_t|s_t)\hat{\pi}^2(b_t|s_t)\right)$$

$$= \sum_{\tau}R^1(\tau)\mathrm{Pr}_{\mu}^{\pi^1,\hat{\pi}^2}(\tau)\sum_{t=0}^{\infty}\nabla_{\theta^1}\left(\log\pi^1(a_t|s_t) + \log\hat{\pi}^2(b_t|s_t)\right)$$

$$= \mathbb{E}_{\tau \sim \mathrm{Pr}_{\mu}^{\pi^1,\hat{\pi}^2}}\left[R^1(\tau)\sum_{t=0}^{\infty}\nabla_{\theta^1}\left(\log\pi^1(a_t|s_t) + \log\hat{\pi}^2(b_t|s_t)\right)\right]$$

$$= \mathbb{E}_{\tau \sim \mathrm{Pr}_{\mu}^{\pi^1,\hat{\pi}^2}}\left[\sum_{t=0}^{T}A^1(s_t,a_t,b_t)\nabla_{\theta^1}\left(\log\pi^1(a_t|s_t) + \log\hat{\pi}^2(b_t|s_t)\right)\right]$$

In reality, we use trajectories sampled from $\pi^1$ and $\pi^2$, to compute our own estimate of the actor-critic update so we have

$$\nabla_{\theta^1}\hat{V}^1(\mu) = \mathbb{E}_{\tau \sim \mathrm{Pr}_{\mu}^{\pi^1,\pi^2}}\left[\sum_{t=0}^{T}A^1(s_t,a_t,b_t)\nabla_{\theta^1}\left(\log\pi^1(a_t|s_t) + \log\hat{\pi}^2(b_t|s_t)\right)\right]$$

## F  REPRODUCIBILITY

For reproducing our results on the IPD and the Coin Game please visit this link. This is an anonymized repository and the instructions for reproducing the results and the seeds are provided. The seeds we used are $42$ to $51$.

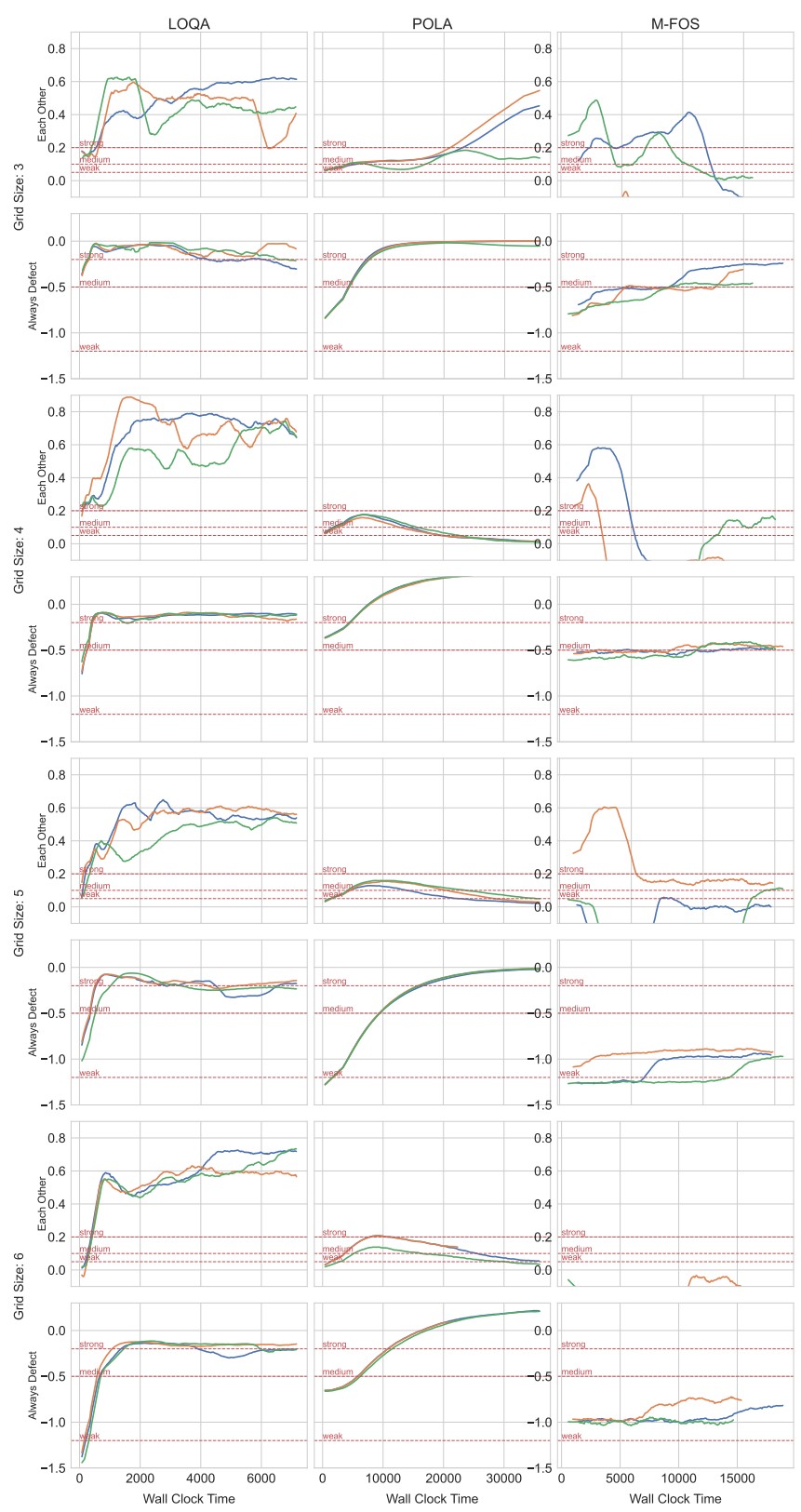

Figure 6: Training curves for 3 seeds of POLA and LOQA for 3x3, 4x4, 5x5 and 6x6 grid sizes on the two metrics in the Coin Game: Normalized return vs. each other and vs. always defect (AD). The wall clock time is measured in seconds. Note that the range of the x-axis is different for POLA, LOQA, and M-FOS as POLA and M-FOS take longer to pass the considered thresholds.

