# OpenReview forum: "LOQA: Learning with Opponent Q-Learning Awareness"
_ICLR.cc/2024/Conference — ICLR 2024 poster_

### Official Review · Reviewer_iBDc · 2023-10-18

**Soundness:** 3 good
**Presentation:** 3 good
**Contribution:** 2 fair
**Rating:** 3
**Confidence:** 4

**Summary:**

the paper presents LOQA, a decentralized reinforcement learning algorithm that optimizes individual utility while promoting cooperation among adversaries in partially competitive environments. LOQA is designed to achieve state-of-the-art performance in benchmark scenarios like the Iterated Prisoner's Dilemma and the Coin Game, making it a promising approach for practical multi-agent applications. The paper provides a detailed description of the LOQA algorithm, including its opponent Q-learning awareness assumption, and presents experimental results that demonstrate its effectiveness.

**Strengths:**

Quality: The paper provides a detailed description of the LOQA algorithm, including its opponent Q-learning awareness assumption, and presents experimental results that demonstrate its effectiveness. The experiments are well-designed and the results are statistically significant.

**Weaknesses:**

the LOQA algorithm assumes the opponent acts accordingly to an inner action-value function and is designed for environments with discrete action spaces. This means that LOQA is unable to shape other opponents that do not necessarily follow this assumption. The assumptions and dependencies of algorithms are strong, making it difficult to handle continuous action space problems and multi-agent issues.

**Questions:**

Q1: The IPD and coingame mentioned in the paper mainly refer to two general-sum games. Can the algorithm be applied to other types of environments, such as zero-sum games, or more complex gaming environments, such as StarCraft II?

---

### Official Review · Reviewer_SYs7 · 2023-10-30

**Soundness:** 2 fair
**Presentation:** 3 good
**Contribution:** 3 good
**Rating:** 6
**Confidence:** 3

**Summary:**

This paper presents a method for decentralized learning in general-sum games, building on the assumption that adversaries sample actions according to their action-value function. The experiments demonstrate significant improvements in wall-clock time and performance against POLA in the coin game.

I am recommending borderline acceptance for this work, as the method is a useful contribution towards scaling LOLA-based methods. However, I would like to see the evaluation strengthened with more and increasingly complex environments, as well as more than one baseline (particularly M-FOS).

**Strengths:**

1. The related work is detailed, covering LOLA and meta-learning approaches to social dilemmas.
2. The method is clearly motivated and described. The underlying assumption is well explained and plausible.
3. The evaluation demonstrates a significant improvement in computational efficiency against POLA, with the analysis demonstrated in figures 3 and 4 showing this scaling behavior well.
4. Figure 1 presents a validation that LOQA is capable of learning tit-for-tat-like strategies in IPD.

**Weaknesses:**

1. The evaluation is limited in diversity, comparing LOQA to a single baseline algorithm (POLA) on a single environment (coin game). The selection of POLA vs alternative LOLA extensions is justified, but I am unsure how M-FOS is neglected when it has also been demonstrated to be effective at the coin game? Furthermore, the choice to only evaluate against POLA on the coin game is limiting, when full-history IPD and chicken would be equally interesting and strengthen the results.
2. The scaling performance of LOQA would be better demonstrated by including further, more complex environments, rather than just scaling up the grid size of the coin game. Since previous LOLA extensions have been prohibitively expensive, this has not been possible, but it appears LOQA runs in a very reasonable amount of time on the largest of these tasks. This gives the opportunity to demonstrate LOQA on a "more complex and realistic scenario", rather than just extrapolating from its scaling performance here.
3. Whilst the training dynamics of POLA are consistent across seeds, LOQA seems to have significant variance (Figure 4). The number of LOQA seeds should be increased beyond 3 to handle this variance.

**Questions:**

1. Nitpick: The scaling of LOQA in figure 3 would be clearer with a log-scale y-axis.

---

> ### Author Response · Authors · 2023-11-11
>
> We would like to thank the reviewer for their time and thoughtful feedback.
>
> **Weaknesses:**
>
> 1. We agree with the reviewer in all of these points. In particular, we will incorporate M-FOS as a baseline for all of our experiments and add full history IPD and Chicken. We did not incorporate M-FOS initially because according to Figure 7 of the M-FOS paper the solution achieved is marginally better than a naive (PPO) agent.
>
> 2. This is a valid observation, but we believe that it holds LOQA to a higher standard than POLA or M-FOS. It also makes potential results difficult to interpret since more complex environments lack strong baselines, therefore putting an extra burden for us in terms of adapting and extending existing methods.
>
> 3. We will also run more LOQA seeds since we believe it is reasonable and doable before the rebuttal deadline.
>
> **Questions:**
>
> 1. We will try a variation of Figure 3 using a log scale and keep it if it still clearly showcases the results.
>
> Provided that we make all of the changes promised above, would you be willing to increase your score?

---

> > ### Comment · Reviewer_SYs7 · 2023-11-19
> >
> > Thank you for your rebuttal. In summary, I am currently maintaining my score pending a large revision.
> >
> > > we believe that it holds LOQA to a higher standard than POLA or M-FOS.
> >
> > You are correct about this point. I do not think this is a requirement for the paper to be published, however, it is a reasonable expectation for follow-up work to raise the standard of contribution. Particularly in this case, when computational scaling is a major contribution of the proposed method, it would be strongly encouraging to see it evaluated on a task with meaningfully scaled complexity. If existing methods entirely fail to solve the method - leading to no strong baselines - this should be easy to demonstrate and would only strengthen the evaluation.
> >
> > > We will incorporate M-FOS as a baseline for all of our experiments and add full history IPD and Chicken...
> > > We will also run more LOQA seeds...
> >
> > I appreciate your receptiveness regarding these experiments, which I believe would strengthen the paper significantly. I would be surprised if these results were achievable in the rebuttal period alone. Assuming they are positive, the paper would also require restructuring and rewriting to include said results.
> >
> > If all of this was achieved before the end of the rebuttal period, I would be inclined to increase my score. However, I suspect this will be a stretch and risks making the paper incoherent if rushed. Regardless, including these results in a future revision would lead to a strong submission, which I encourage you to pursue.

---

### Official Review · Reviewer_rBSb · 2023-10-31

**Soundness:** 3 good
**Presentation:** 3 good
**Contribution:** 2 fair
**Rating:** 3
**Confidence:** 4

**Summary:**

The paper proposes LOQA -- a new member of the LOLA family (next to COLA and POLA). Conceptually, LOQA is different in that it assumes that the opponent uses Q-leaning (more specifically, that the probability of its action in a state is proportional to the corresponding Q-value relative to Q-values of other actions). The advantage of this algorithm is that it can achieve performance comparable to POLA (previous state-of-the-art of the family) but faster, which also improves scalability to bigger environments. This claim is complemented by experiments in the Coin Game, which is a two-agent coin collection game on a small grid with 2 agents.

**Strengths:**

- Solutions to conditional (equilibrium-based) cooperation in general and improvements of LOLA in particular are relevant to MARL.
- The paper is straightforward and well-written.
- The experiments are sound, I especially like fig. 2.

**Weaknesses:**

- Related work lacks discussion of MARL approaches to learning prosocial equilibria other than reciprocity-based or opponent-shaping-based, such as reward redistribution https://ala2020.vub.ac.be/papers/ALA2020_paper_45.pdf https://arxiv.org/abs/2004.13332, mediation https://arxiv.org/pdf/2306.08419.pdf, contracts https://arxiv.org/pdf/2208.10469.pdf, and similarity-based equilibria https://arxiv.org/pdf/2211.14468.pdf.
- Some limitations of previous LOLA-based approaches that LOQA does not fix are unmentioned in the Limitations section: e.g., it is only applicable to environments with 2 agents.
- The contribution is limited to modifying an existing algorithm and improving its speed, but not performance. The new algorithm is also quite complex. I do not think the contribution is sufficient for ICLR. A workshop would be a better fit.

**Questions:**

- I am not sure why in 5.1 the Q-value $\hat{Q}^2$ is approximated as an empirical return. Since we need access to the real $Q^2$ or its approximation through opponent modeling regardless, can we not use a 1-step or n-step temporal difference instead (which would give a better bias-variance)?

---

> ### Author Response · Authors · 2023-11-11
>
> We are happy that the reviewer finds LOQA to be a scalable method for solving social dilemmas.
>
> **Summary:**
>
> LOQA **is not** an algorithm that can be considered *“a new member of the LOLA family”* as it is fundamentally different. We would like to remind the reviewer that algorithms in the LOLA family differentiate w.r.t. imagined opponent parameter updates that are computed explicitly. This causes significant computational burden, making these algorithms hard to scale. LOQA on the other hand differentiates w.r.t. the opponent’s Q function (via reinforce), which is estimated by observing the returns of trajectories. Therefore LOQA does not compute gradients w.r.t. explicit optimization steps.
>
> **Weaknesses:**
>
> 1. We thank the reviewer for the suggested additions to the relevant literature, in particular we believe that similarity-based equilibria should be discussed. However, we want to emphasize that we are interested in the setup of solving social dilemmas without third parties (reward redistribution,  mediation), contracts, or any other changes to the underlying game.
>
> 2. This statement is not true, and should be clarified further. Both LOLA and LOQA can be extended to games with more than two players. LOQA would do it by adding extra log terms for each extra player to the loss, whereas LOLA would require computing imagined updates for each new player. This is not explicitly done as social dilemmas with two players are already difficult to solve for current algorithms.
>
> 3. We respectfully disagree with the reviewer. LOQA does not belong to the LOLA family and as such is not a modification of an existing algorithm. We feel that LOQA is being held to a higher standard than existing methods in the literature. How is a novel method that improves the state of the art for neural network solutions to social dilemmas (both in terms of optimality and speed) only suited for a workshop, especially considering that previous publications (POLA, M-FOS) have been published to top machine learning venues? Also, we kindly request the reviewer to clarify the definition of 'complexity' in this context as we believe LOQA is relatively simple. In contrast to previous work, LOQA does not have inner outer loops, differentiation through optimization steps, and meta-games. It would be helpful if they could offer arguments to support the views that a) LOQA is more complex compared to existing methods, and b) why this kind of complexity might be considered undesirable.
>
> **Questions:**
>
> 1. We experimented with bootstrapped versions of the $\hat{Q}^2$ estimate for many different lengths initially with worse results than using the full trajectory. We believe that, since the only differentiable parts of the estimates are the rewards, making the full trajectory differentiable works better for longer games.
>
> We are surprised by the suggestion that our paper is better suited for a workshop. Given that we have addressed the reviewer’s concerns and criticism, we ask the reviewer to reconsider their score.

---

> > ### Comment · Reviewer_rBSb · 2023-11-13
> > **Response to authors**
> >
> > The authors' rebuttal has not provided much new information and instead emphasizes points already made in the paper (which is somewhat useful) and channels the authors' frustration and subjective disagreement (which is not useful). Still, I will clarify my assessment so that authors can improve their manuscript.
> >
> > **Summary**
> >
> > This is a semantic argument and thus is unconstructive to argue about.
> >
> > **Weaknesses**
> >
> > The assessment that much of the literature I provided should not even discussed as a part of related work is obviously inadequate. Notice that I do not propose to compare with these methods, as I agree with the authors' points about those methods changing the game / introducing a third party, unlike LOQA. Still, they solve the same problem, and unlike LOQA, are applicable to n-player games, some of them even to MeltingPot (that is much more complex than the coin game). So whichever limitations they may have, they have advantages over LOQA. Both should be discussed in an adequate literature review.
> >
> > Regarding the n-player games, the authors claim in the rebuttal that their algorithm is applicable, but to my understanding, the paper does not discuss it and certainly does not demonstrate it. There's a limit to how much the argument that LOQA is held to a higher standard can justify. N-player does not equal MeltingPot. In the literature that the authors so readily dismissed, there already exist examples of sequential n-player games with high-dimensional but relatively simple state spaces, which could match the LOQA's higher standard. There is no reason to not demonstrate LOQA's applicability to these games, which in my opinion would significantly strengthen the manuscript (as the previous algorithms in the LOLA family have not been demonstrated in n-player games).
> >
> > My assessment of incremental contribution stands. However more reasonable the method may seem to authors than predecessors, the general approach is the same, and the empirical results are not impressive enough.
> >
> > **Questions**
> >
> > Thank you for the clarification, makes sense to me.

---

> > > ### Author Response · Authors · 2023-11-13
> > >
> > > We thank the reviewer for their promptness and willingness to contribute to the discussion. We would like to emphasize that the Coin Game is actually a challenging environment in the context of reinforcement learning for social dilemmas since it requires a neural network parameterization of the policy. The difficulty of the problem is also tied to the non-stationary nature of the environment and the delays in reward. This is supported by the fact that very few algorithms in the literature succeed in the Coin Game.
> > >
> > > **Summary**
> > >
> > > We do not believe this is unconstructive because the reviewer originally stated “*The contribution is limited to modifying an existing algorithm and improving its speed, but not performance.*”. Our clarification is meant to disprove the claim that LOQA is a modification to an existing algorithm. In that regard we believe that the reviewer now agrees that a) LOQA is not a modification of an existing algorithm and b) LOQA improves both speed and performance (measured by Pareto optimality in the Coin Game and demonstrated in the experiments) when compared to existing methods.
> > >
> > > **Weaknesses**
> > >
> > > 1. We believe that this is a valid observation and agree with the reviewer that these other methods should be included in the literature review, so we will include them.
> > >
> > > 2. Our motivation with this argument is to bring the reviewer to clearly state which are the limitations that they believe “*previous LOLA-based approaches that LOQA does not fix are unmentioned in the Limitations section*”. As of now, these limitations have not been stated. We agree with the reviewer that there is room for experimental improvement either by using other environments or by increasing the number of players. We believe we could perform an experiment in which we compare LOQA with baselines in a 3-player version of the Coin Game (before the rebuttal period is over) and ask the reviewer if this would help clarify their questions regarding n-player games and strengthen the results.
> > >
> > > 3. When we state that LOQA is being held to a higher standard than their predecessors we just intend to underline that LOQA demonstrates state-of-the-art performance in a challenging environment, i.e. the Coin Game. Therefore we would like the reviewer to justify why “*the general approach is the same, and the empirical results are not impressive enough*”, when our method is novel and demonstrates state of the art performance in the existing baselines. Understanding this assessment could be very helpful for us to improve the paper.

---

> > > > ### Comment · Reviewer_rBSb · 2023-11-16
> > > >
> > > > Providing results in the Coin Dilemma with 3 agents would improve the contribution, but I'm unsure if it will be sufficient. I suggest the authors take the time to do a thorough experimental investigation and try even more agents, rather than rushing during the rebuttal period.

---

### Official Review · Reviewer_xNg3 · 2023-11-01

**Soundness:** 2 fair
**Presentation:** 2 fair
**Contribution:** 2 fair
**Rating:** 5
**Confidence:** 3

**Summary:**

This paper introduces learning with opponent Q-learning awareness (LOQA) which optimizes cooperation in mixed-motive environments by assuming the opponent samples actions proportionally to Q values. This method is computationally lighter compared with prior works.

**Strengths:**

1. I like the idea of deriving some cooperative solutions in mixed-motive games without computing the meta-game solutions. There were many efforts in this direction but only a few paid off, the main limitation lies in the scalability of the multi-agent problems.
2. The paper is clear and easy to follow.

**Weaknesses:**

In general, the experimental part has room for improvement
1. When this line of research on LOLA has a few prior works, a comparison with a decent amount of previous works is necessary so that we know the proposed method is better. The good performance of a particular method under a particular environment is not the reason to abandon other methods, especially when POLA [1] did not compare with M-FOS [2]
2. Results on the IPD and coin environment may be a bit preliminary when we jointly consider the contribution of the algorithm (efficiency). More complex games like Meltingpot 2.0 [3] with some clearly diverse background agent policies (cooperating for different variations of time and defect afterward) can be a good benchmark for the completeness of the experiments

[1] Zhao, S., Lu, C., Grosse, R. B., & Foerster, J. (2022). Proximal Learning With Opponent-Learning Awareness. Advances in Neural Information Processing Systems, 35, 26324-26336.

[2] Lu, C., Willi, T., De Witt, C. A. S., & Foerster, J. (2022, June). Model-free opponent shaping. In International Conference on Machine Learning (pp. 14398-14411). PMLR.

[3] Agapiou, J. P., Vezhnevets, A. S., Duéñez-Guzmán, E. A., Matyas, J., Mao, Y., Sunehag, P., ... & Leibo, J. Z. (2022). Melting Pot 2.0. arXiv preprint arXiv:2211.13746.

**Questions:**

1. (Comments) The authors should use \citep{} for (Author, Year) citations
2. Any intuitions on why the other work[1], meta-game + model-free opponent-shaping does not work on coin game?

[1] Lu, C., Willi, T., De Witt, C. A. S., & Foerster, J. (2022, June). Model-free opponent shaping. In International Conference on Machine Learning (pp. 14398-14411). PMLR.

---

> ### Author Response · Authors · 2023-11-11
>
> We would like to thank the reviewer for their thoughtful feedback.
>
> **Weaknesses:**
>
> 1. First, we want to restate that there are not many existing methods that work in the Coin Game. That being said, we are only aware of POLA and M-FOS; LOLA has been shown to not work by the POLA authors. We will add an M-FOS comparison as we believe it is a valid and reasonable concern. We did not do so initially because we had reason to believe that the M-FOS results were marginally better than those of naive agents. We refer the reviewer to Figure 7. in the M-FOS paper, where the odds of taking their own coin reach 0.54 which demonstrates weak cooperation compared to LOQA and POLA.
>
> 2. We agree that running LOQA in more complex environments like Meltingpot 2.0 would strengthen the experimental part of the paper, but LOQA is being held to a much higher standard than the existing papers in the literature. In LOLA, M-FOS and POLA, the environments are limited to IPD, the Coin Game and some other matrix form games, but we are being asked to run experiments in environments for which baselines do not exist.  Consider that running these experiments with LOQA and other baseline methods requires significant engineering effort.
>
> We will also add experiments for full history IPD and Chicken. Assuming that we perform these changes and add M-FOS as a baseline, would you be willing to increase your score?

---

> > ### Comment · Reviewer_xNg3 · 2023-11-20
> >
> > My first impression of this paper is it was written in a rush. However, if the work has been improved over the months and the authors would update their manuscripts during the rebuttal, this effort should be valued and I would be happy to read it again.
> >
> > Please highlight the changes with different colors. Please note that "increasing scores" is not guaranteed and the scores will based on the updated version of the paper.

---

### Official Review · Reviewer_F4Df · 2023-11-01

**Soundness:** 2 fair
**Presentation:** 2 fair
**Contribution:** 2 fair
**Rating:** 3
**Confidence:** 3

**Summary:**

In this paper, the authors propose a decentralized multi-agent learning algorithm that fosters cooperation among agents even in adversarial settings, which they term as partially competitive environments. They provide experimental validation with Iterated Prisoners' Dilemma and Coin Game. Their key claim is that their proposed algorithm achieves state-of-the-art performance in these games with low computational cost. The authors here assume that each agent has access to the Q values of all other agents or can estimate them using the observations and rewards of all other agents.

**Strengths:**

The authors try to address the computational challenges faced by other MARL algorithms for sequential social dilemmas, by proposing an algorithm where each agent maintains an estimate of the Q values of all its opponents in order to determine its own policy improvement.

**Weaknesses:**

1. There are several papers in literature that provide decentralized algorithms to achieve individually and socially optimal solution in sequential social dilemmas. One of the criticism of these papers is the additional information needed by these algorithms, which is often not available in the real-world. This paper also has the same limitations.
2. I think the novelty in the proposed method is limited based on the papers cited by it. The key idea is that each agent model the opponents policy using the rewards obtained by the opponents.

**Questions:**

1. Can the authors concretely define "partially competitive settings"?
2. What is the information structure assumed for each agent? If each agent can see the entire world state and also observe the actions of the other agent, then the policy for each agent should also depend on the history of the actions of the agents in addition to the current state for no loss of optimality.
3. How can the opponent's true Q function be replaces with an estimate by the agent? Are we assuming that opponent observations and rewards are common information in the game? This is often not the case in real-worls multi-agent settings.
4. It is not clear if LOQA will extend to all general-sum games. Also, the sub-optimality for each agent due to the modified objective function needs to be quantified and bounded. Are the authors proposing LOQA only for sequential social dilemmas?
5. Can the authors establish or reason about the solution concept achieved using LOQA? Will it be a socially optimal solution. In non-symmetric games what will be the extent of sub-optimality for each agent?
6. By modifying the objective function (returns) optimized by each agent, LOQA changes the underlying game. Can the authors show that a Nash equilibrium (or any refinement of it) of this modified game is a socially optimal solution of the original game?
7. Also, will decentralized REINFORCE algorithm lead to attaining a Nash equilibrium in the above-defined modified game?
8. How does this analysis extend to the case when all agents in a game are using LOQA?

---

> ### Author Response · Authors · 2023-11-11
>
> **Strengths:**
>
> We thank the reviewer for their time. The reviewer states that the strength of the paper lies in *“proposing  an algorithm where each agent maintains an estimate of the Q values of all its opponents in order to determine its own policy improvement”*. We would like to clarify that estimating the Q values of the opponents is not enough to determine a policy improvement and is not a contribution of our work. The key idea of LOQA is that this estimate of the Q value is controllable by the policy of the agent: a LOQA agent exercises this control over the Q values of the opponent to incentivize it to select actions that are beneficial for it.
>
> **Weaknesses:**
>
> 1. The reviewer writes that *“There are several papers in literature that provide decentralized algorithms to achieve individually and socially optimal solution in sequential social dilemmas”*. Besides those that we have included in the literature review we are not aware of the existence of such papers, we would like the reviewer to point them out. For those in the literature review we have clearly enunciated their differences and disadvantages in comparison to LOQA.
>
> 2. We believe that the reviewer may have missed the key idea of the paper, namely that the Q value of the opponent can be differentiated w.r.t. the agent’s policy parameters via reinforce. In such a way, it is a fundamentally different algorithm from those that we cite in the paper. This idea enables major computational savings. In contrast to previous works which compute explicit gradients w.r.t to optimization steps or model the problem as a meta-game (both of which are computationally expensive), LOQA uses reinforce estimators that are fast and easy to compute.
>
> **Questions**
>
> 1. We agree with the reviewer that "partially competitive settings" is not clearly defined. We will replace this term with “social dilemmas”.
>
> 2. Our policies in the Coin Game are GRU policies that aggregate the entire observation history (without actions) up to the current time step through the hidden state propagation. Therefore, the agents condition on the entire observation history.
>
> 3. Yes, we assume that the rewards of both agents can be observed by both agents. This might not be the case in many settings, but we believe it is a reasonable assumption as previous works also make it.
>
> 4. The reviewer’s concern that LOQA may not generalize to all general sum games is valid, but we intended to demonstrate its usefulness only in social dilemmas.
>
> 5. LOQA achieves a Nash equilibrium (tit-for-tat) on one-step history IPD as demonstrated by the experimental results. Beyond that it is difficult to formally characterize the solutions found by LOQA and it is not the scope of the paper.
>
> 6. LOQA does not modify the returns optimized by each agent, so this statement is not true.
>
> 7. As stated in (6), there is no modification to the underlying game, thus we will not address this question.
>
> 8. Then again refer to (6).
>
> We ask the reviewer to increase their score.

---

> > ### Comment · Reviewer_F4Df · 2023-11-22
> >
> > I thank the authors for their responses. On reading these responses and the comments by the other reviewers, I think I will retain my current score.

---

### Meta-Review · Area_Chair_nPLp · 2023-12-07

**Metareview:**

This paper introduces a novel approach towards opponent shaping.
Specifically, the policy of the other agents is modelled as a softmax policy over Q-values of the other agent.
Crucially, these Q-values directly depend on the action selection of the first agent and thus can be differentiated wrt the weights of said first agent. This insight, in combination with recent progress in stochastic gradient estimation, allows the method to differentiate through the opponents policy to do opponent shaping extremely efficiently and stably.

In my personal assessment, this paper is most certainly not a clear reject. I believe most of the concerns of the reviewers were either addressed by the authors or are due to misunderstanding by the reviewers.
The paper is also clearly written, extremely reproducible (due to published code) and the method is innovative.

The only relevant remaining concern that I could identify is the lack of comparison to recent approaches such as Model-Free Opponent Shaping. The authors stated that they would run the experiment but did not provided any updated results.

However, I do not think that this is sufficient reason for rejecting the paper and vote for acceptance, under the condition that the authors add the missing comparisons for the camera ready copy.

**Justification For Why Not Higher Score:**

NA

**Justification For Why Not Lower Score:**

NA

---

### Decision · Program_Chairs · 2024-01-16

Accept (poster)